# Empirical Studies on the Properties of Linear Regions in Deep Neural Networks

**Xiao Zhang & Dongrui Wu**[*]
Ministry of Education Key Laboratory of Image Processing and Intelligent Control,
School of Artificial Intelligence and Automation,
Huazhong University of Science and Technology, Wuhan, China
`{xiao_zhang,drwu}@hust.edu.cn`

## Abstract

A deep neural network (DNN) with piecewise linear activations can partition the input space into numerous small linear regions, where different linear functions are fitted. It is believed that the number of these regions represents the expressivity of the DNN. This paper provides a novel and meticulous perspective to look into DNNs: Instead of just counting the number of the linear regions, we study their local properties, such as the inspheres, the directions of the corresponding hyperplanes, the decision boundaries, and the relevance of the surrounding regions. We empirically observed that different optimization techniques lead to completely different linear regions, even though they result in similar classification accuracies. We hope our study can inspire the design of novel optimization techniques, and help discover and analyze the behaviors of DNNs.

## 1 Introduction

In the past few decades, deep neural networks (DNNs) have achieved remarkable success in various difficult tasks of machine learning (Krizhevsky et al., 2012; Graves et al., 2013; Goodfellow et al., 2014; He et al., 2016; Silver et al., 2017; Devlin et al., 2019). Albeit the great progress DNNs have made, there are still many problems which have not been thoroughly studied, such as the expressivity and optimization of DNNs.

High expressivity is believed to be one of the most important reasons for the success of DNNs. It is well known that a standard deep feedforward network with piecewise linear activations can partition the input space into many linear regions, where different linear functions are fitted (Pascanu et al., 2014; Montufar et al., 2014). More specifically, the activation states are in one-to-one correspondence with the linear regions, i.e., all points in the same linear region activate the same nodes of the DNN, and hence the hidden layers serve as a series of affine transformations of these points. As approximating a complex curvature usually requires many linear regions (Poole et al., 2016), the expressivity of a DNN is highly relevant to the number of the linear regions.

Studies have shown that the number of the linear regions increases more quickly with the depth of the DNN than with the width (Montufar et al., 2014; Poole et al., 2016; Arora et al., 2018). Serra et al. (2018) detailed the trade-off between the depth and the width, which depends on the number of neurons and the size of the input. However, a deep network usually leads to difficulties in optimization, such as the vanishing/exploding gradient problem (Bengio et al., 1994; Hochreiter, 1998) and the shattered gradients problem (Balduzzi et al., 2017). Batch normalization (BN) can alleviate these by repeatedly normalizing the outputs to zero-mean and unit standard deviation, so that the scale of the weights can no longer affect the gradients through the layers (Ioffe & Szegedy, 2015). Another difficulty is that the high complexity caused by the depth can easily result in overfitting. Srivastava et al. (2014) proposed dropout to reduce overfitting, which allows a DNN to randomly drop some nodes during training and work like an ensemble of several thin networks during testing.

Despite the empirical benefits of these techniques, their effects on the trained model and the reasons behind their success are still unclear. Previous studies focused on explaining why these techniques

---

[*]Dongrui Wu is the corresponding author.

can help the optimization during training (Wager et al., 2013; Santurkar et al., 2018; Bjorck et al., 2018). Different from theirs, our study is trying to answer the following question:

*What properties do these techniques introduce to the linear regions after training?*

We present an intuitive view of different techniques' effects on the linear regions in Figure 1. We may observe that:

1. BN and dropout help the DNN partition the input space into many more linear regions than the vanilla DNN.
2. The linear regions resulted from the BN model are more uniform in size than those from the dropout DNN.
3. Many transition boundaries of the dropout DNN share similar norm directions, and concentrate around the decision boundaries.

Figure 1 illustrates that different optimization techniques can lead to completely different linear regions even using the same DNN architecture, which may influence the behaviors of the DNN, such as its adversarial robustness (Biggio et al., 2013; Szegedy et al., 2014) or some other undiscovered ones (Appendix A shows another example on a two-dimensional toy dataset). Therefore, it is important to probe the properties of the linear regions introduced by these frequently-used optimization techniques, instead of just looking at the learning curves.

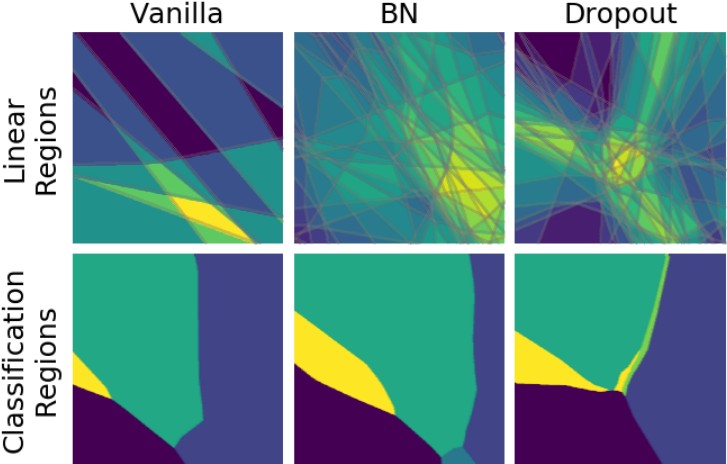

Figure 1: Linear regions and classification regions of models trained with different optimization techniques. The gray curves in the top row are transition boundaries separating different linear regions, and the color represents the activation rate of the corresponding linear region. In the bottom row, different colors represent different classification regions, separated by the decision boundaries. The models were trained on the vectorized MNIST dataset, and this figure shows a two-dimensional slice of the input space. The vanilla model was a fully-connected ReLU network with three hidden layers. To make the figure more readable, each hidden layer only included 20 nodes. The BN model was the vanilla model with BN added after pre-activations of every hidden layer, and the dropout model added dropout layers after the hidden layers, with drop rate 0.2.

This paper introduces mathematical tools of polyhedral computation into linear region analysis, and systematically compares various properties of the linear regions after DNNs being trained with different optimization techniques (BN, dropout). Though the models have similar classification accuracies, we do observe significant differences among their linear regions:

1. *BN can help DNNs partition the input space into smaller linear regions, whereas dropout helps around the decision boundaries.*
2. *BN can make the norm directions of the hyperplanes orthogonal, whereas dropout the opposite.*

3. *Dropout makes regions in which data points lie less likely to contain the decision boundaries.*

4. *The gradient information of a linear region has high relevance to its surrounding regions; however, BN slightly reduces the relevance.*

It should be noted that our approach can also be applied to analyzing the preferences introduced by other optimization techniques, such as different initialization approaches (Glorot & Bengio, 2010; He et al., 2015), different optimizers (Duchi et al., 2011; Kingma & Ba, 2014), some widely-used operators like skip-connection (He et al., 2016), and even different hyperparameters. Since training deep learning models is related to the complex interaction among the training dynamics, data manifold and model architectures, we believe it is better to decouple it into two subtasks: 1) *figure out what kinds of linear regions can best approximate the data manifold*; and, 2) *design optimization techniques to achieve such linear regions*. Our work provides tools to analyze the linear regions, which helps study both subtasks.

The remainder of the paper is organized as follows: Section 2 presents the details of the linear regions of DNNs with ReLU activation. Section 3 shows how BN and dropout influence the linear regions. Section 4 draws conclusions.

## 2 LINEAR REGIONS

This section describes the detailed approach[1] for searching for the linear regions of a given input, and introduces some basic concepts of convex polytopes.

### 2.1 SEARCH FOR THE LINEAR REGIONS

A linear region is the intersection of a finite number of halfspaces. Therefore, to search for the linear region in which a given input point lies, we only need to search for its corresponding halfspaces, which are determined by the activation state of the DNN.

Let's consider a vanilla fully-connected DNN with $L$ hidden layers and ReLU activation. Mathematically, let $\mathbf{x} \in \mathbb{R}^d$ denote the $d$-dimensional vectorized input, $\mathbf{h}^l(\mathbf{x})$ the pre-activation outputs of the $l$-th hidden layer with $n_l$ nodes ($l = 1, 2, ..., L$), $\mathbf{z}(\mathbf{x})$ the logits of the output layer, and $M$ the number of classes. Given an input example $\mathbf{x}^*$, because of the one-to-one correspondence relationship between a linear region and an activation state, its corresponding linear region is a set of inputs which lead to the same activation state of the DNN.

Let $S_l$ denote a set of the inputs whose activation states of the first $l$ hidden layers are the same as those of $\mathbf{x}^*$. Obviously, $S_{l+1} \subseteq S_l$. Consider the first layer. Since $\mathbf{h}^1 : \mathbb{R}^d \to \mathbb{R}^{n_1}$ is simply an affine transform of $\mathbf{x}$, according to the activation state, $S_1$ can be written as:

$$S_1 = \left\{ \mathbf{x} \mid \mathbf{w}_i^T \mathbf{x} + b_i \geq 0, \quad \forall i \in \{1, ..., n_1\} \right\}, \tag{1}$$

where

$$\mathbf{w}_i = \text{sgn}(\mathbf{h}_i^1(\mathbf{x}^*)) \nabla_{\mathbf{x}} \mathbf{h}_i^1(\mathbf{x}^*), \tag{2}$$

$$b_i = \text{sgn}(\mathbf{h}_i^1(\mathbf{x}^*)) \left[ \mathbf{h}_i^1(\mathbf{x}^*) - (\nabla_{\mathbf{x}} \mathbf{h}_i^1(\mathbf{x}^*))^T \mathbf{x}^* \right]. \tag{3}$$

Next, we select the points from $S_1$ to construct $S_2$, which activate the same nodes of the second layer as $\mathbf{x}^*$ does. Recall that the points in $S_1$ lead to the same activation state of the first layer, hence $\mathbf{h}^2 : S_1 \to \mathbb{R}^{n_2}$ is also an affine transform of $\mathbf{x}$. Therefore, $S_2$ can be constructed by adding linear inequality constraints to $S_1$, in the same way as (1). Due to the linearity of $\mathbf{h}^l : S_{l-1} \to \mathbb{R}^{n_l}$ (where $S_0 = \mathbb{R}^d$) for all $l$, this procedure can be repeated till the last hidden layer. The detailed deduction is presented in Appendix B, and the depth-wise exclusion process to search for a linear region of $\mathbf{x}^*$ is illustrated in Figure 2.

---

[1]We developed the approach for finding the linear regions independently, but later found that the same approach has been proposed in (Lee et al., 2019). However, the two papers use the linear regions for different purposes.

Let $\mathcal{C}^* = \{(\mathbf{w}_i, b_i)\}_{i=1}^{\sum_{l=1}^{L} n_l}$ be the set of parameters of all the linear inequality constraints. Then $S_L$, which is the linear region that $\mathbf{x}^*$ lies in, can be represented as the intersection of $\sum_{l=1}^{L} n_l$ halfspaces:

$$S_L = \left\{ \mathbf{x} \mid \mathbf{w}_i^T \mathbf{x} + b_i \geq 0, \quad \forall (\mathbf{w}_i, b_i) \in \mathcal{C}^* \right\}, \tag{4}$$

where the DNN is a locally linear model, as $\mathbf{z} : S_L \to \mathbb{R}^M$ is an affine transform of $\mathbf{x}$.

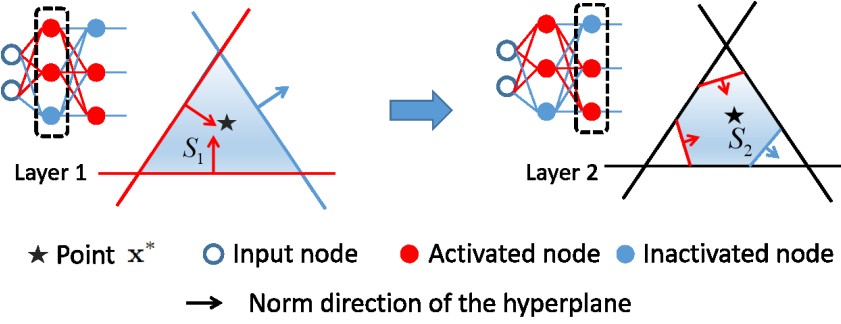

Figure 2: Depth-wise exclusion process to search for the linear region of $\mathbf{x}^*$. Each node in each layer provides a hyperplane to cut the polytope defined by the preceding layers.

Though we mainly discuss the standard fully-connected DNNs, the form to describe a linear region can be extended to other situations with little modification. For instance, BN is just a reparameterization of the weights, a convolutional neural network (CNN) can be viewed as a sparse fully-connected DNN, and max-pooling is simply adding more linear inequalities to identify the maximum.

## 2.2 CONVEX POLYTOPES

A linear region can be represented as the set of solutions to a finite set of linear inequalities in (4), which is exactly the *H-representation* of a convex polyhedron. With the natural bounds of the input value, these linear regions are convex polytopes[2]. We omit the word *convex* in the rest of the paper to avoid repetition.

A polytope can also be represented as a convex hull of a finite set of points, which is known as the *V-representation*. Different representations may lead to different complexities when dealing with the same problem; nevertheless, it is challenging to convert one representation into the other (Toth et al., 2017). As the dimensionality increases, it is even harder to calculate the number of vertices/facets with a single H-/V- representation (Linial, 1986), or to verify the equivalence between an H-representation and a V-representation (Freund & Orlin, 1985). More information about polyhedral computation can be found here[3].

It should be noted that convex optimization is capable of probing the properties of the linear regions, because H-representation is a natural description of a convex feasible region, and a DNN's behavior is completely linear in this feasible region.

## 3 PROPERTIES OF THE LINEAR REGIONS

This section presents various properties of the linear regions of DNNs, which were trained using different optimization settings.

### 3.1 MODELS

We used a fully-connected DNN as our vanilla model, which consisted of three hidden layers with 1,024 rectified linear nodes in each. The BN model was the vanilla model with BN added after the

---

[2]Here we use the term *polytope* to denote a bounded polyhedron.
[3]https://inf.ethz.ch/personal/fukudak/polyfaq/polyfaq.html

pre-activations of each hidden layer, and the dropout model added dropout after the activations. We trained our models on the vectorized MNIST dataset, which were rescaled to $[-1, 1]$.

All models were trained using Adam optimizer (Kingma & Ba, 2014) with the default setting ($\beta_1 = 0.9$, $\beta_2 = 0.999$) but different learning rates (1e-3 and 1e-4), and the batch size was 256. We used the Xavier uniform initializer for weight initialization, with biases set to zero (Glorot & Bengio, 2010). Early stopping was used to reduce overfitting. The classification accuracies are shown in Table 1. Since it is unlikely to explore the influence of all optimization settings, this paper emphasizes two optimization techniques, i.e., BN and dropout, and compares the influence of different learning rates.

Table 1: Test accuracies (%) of different DNN models.

| Learning Rate | Vanilla | BN | Dropout |
|:---:|:---:|:---:|:---:|
| 1e-3 | 97.8 | 97.9 | 97.5 |
| 1e-4 | 98.0 | 98.3 | 98.2 |

As shown in Section 2, the number of constraints in (4) equals the number of hidden nodes; hence, it is hard to handle the constraints as the scale of DNN increases, such as removing the redundant inequalities. In our experiments, the number of constraints was $1,024 \times 3 = 3,072$. In addition to the natural bounds for the $28 \times 28 = 784$ pixel values ($\text{MIN}_x \leq \mathbf{x} \leq \text{MAX}_x$), each linear region is defined by $3,072 + 784 \times 2 = 4,640$ linear inequalities.

Our analysis of CNNs trained on the CIFAR-10 dataset is presented in Appendix E.

## 3.2  THE INSPHERES

We first probed the inspheres of the linear regions, which are highly related to the expressivity of a DNN. Though we cannot make a definite statement that a small insphere always leads to a small linear region, it does indicate a region's narrowness.

The insphere of a polytope can be found by solving the following convex optimization problem:

$$\max_{\overline{\mathbf{x}}, r} \quad r \tag{5}$$
$$\text{s. t.} \quad \mathbf{w}_i^T \overline{\mathbf{x}} - r\|\mathbf{w}_i\| + b_i \geq 0, \quad \forall (\mathbf{w}_i, b_i) \in \mathcal{C}^*,$$
$$\text{MIN}_x + r \leq \overline{\mathbf{x}} \leq \text{MAX}_x - r,$$

where $\overline{\mathbf{x}}$ is the center of the insphere, $r$ the inradius, and $\mathcal{C}^*$ the set of parameters of $S_L$ defined in (4). The main idea of the optimization is to find a point in a polytope, whose distance to the nearest facet is the largest.

As shown in Figure 1, linear regions vary according to their closeness to the decision boundaries, hence we studied the inspheres of three different categories of regions[4], which are defined as follows:

**Manifold region:** The region in which the test points lie.

**Decision region:** The region which contains the *normal decision boundaries*. We define a normal decision boundary as the classification boundary between one test point and the mean point of another class. When we test our models, the mean point of a class is classified into the same class for the MNIST dataset.

**Adversarial region:** The region which contains the *adversarial decision boundaries*. We define the adversarial decision boundary as the classification boundary between a test point and its adversarial example. The adversarial examples were generated by the projected gradient descent (PGD) method (Madry et al., 2018).

Linear interpolation was used to search for regions which contain the decision boundaries. Let $\mathbf{x}_{test}$ denote the test point, and $\mathbf{x}_{target}$ the adversarial point or the mean point of another class. We searched for the maximum $\alpha \in [0, 1]$ which causes $\mathbf{x}_\alpha = \alpha \cdot \mathbf{x}_{target} + (1 - \alpha) \cdot \mathbf{x}_{test}$ to be classified into the same class as $\mathbf{x}_{test}$, and the region in which $\mathbf{x}_\alpha$ lies contains the decision boundaries.

---

[4]These regions should not be confused with a *classification region*, which is partitioned according to the classes of the points in the input space.

As shown in Figure 3, different optimization settings introduce significantly different inradius distributions. Compared with the vanilla models, BN leads to the smallest inradiuses of the linear regions, whereas dropout, as shown in the second column of Figure 3, makes the decision regions narrower. It also shows that the inradius distributions of the adversarial regions are more similar to those of the manifold regions, which demonstrates the fundamental differences between the adversarial and the normal decision boundaries. Comparing the two rows in Figure 3, we can observe that a larger learning rate increases the variance of the distributions. However, BN seems to make the inradius distribution less sensitive to the learning rate, which is consistent with the observations in Bjorck et al. (2018) that BN enables training with larger learning rates. Comparing the first two columns of Figure 3, we can also find that the decision regions are usually narrower than the manifold regions, which was also observed by Novak et al. (2018) that on-manifold regions are larger than off-manifold regions. It is reasonable because decision boundary curves need more sophisticated approximation.

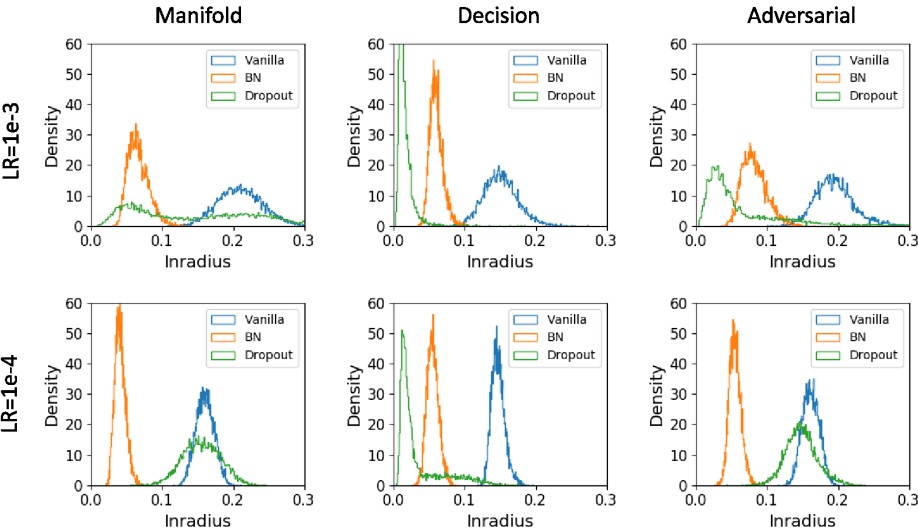

Figure 3: Inradius distributions for different learning rates and optimization techniques. The models in the top row were trained with a larger learning rate (1e-3), and the bottom row a smaller learning rate (1e-4). The three columns show the inradius distributions of the manifold regions, the decision regions, and the adversarial regions, respectively.

### 3.3 DIRECTIONS OF THE HYPERPLANES

Another observation is that BN makes the directions of the hyperplanes of a linear region orthogonal, whereas dropout the opposite. Since it is time-consuming to eliminate the redundant hyperplanes, as explained in Appendix C, we calculated the angles using all hyperplanes.

The direction $\mathbf{w}$ of a hyperplane is pointing to the interior of the linear region, because all constraints are in the form of $\mathbf{w}^T\mathbf{x} + b \geq 0$. Figure 4 shows the angles between directions of every two different hyperplanes of a manifold region. The directions of the hyperplanes become more consistent as the layer goes deeper. Dropout and large learning rates amplify this trend, whereas BN makes the directions more orthogonal. Though we only present one example here, the patterns can be observed for almost all manifold regions.

As shown in (2), the direction $\mathbf{w}$ is actually the gradient of the corresponding hidden node with respect to the input, hence the angle between the directions can be viewed as the correlation between the gradients. However, it should be noted that the shattered gradients problem (Balduzzi et al., 2017) cannot explain the decorrelation characteristics of BN, though they look similar. The shattered gradients problem studies the correlation between the gradients with respect to different inputs, whereas our observation is about the gradients of different hidden nodes with respect to a fixed input.

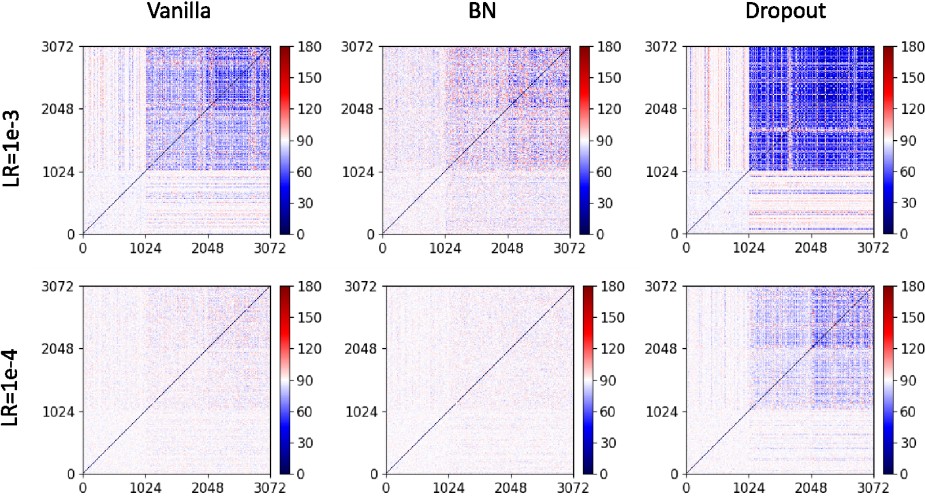

Figure 4: Angles between the directions of different hyperplanes of a particular manifold region. The indices on both axes indicate different hyperplanes: 0-1,023 represent the hyperplanes of the first hidden layer, 1,024-2,047 the second layer, and 2,048-3,071 the third layer. The value of a pixel is $\arccos \frac{\mathbf{w}_i^T \mathbf{w}_j}{\|\mathbf{w}_i\|\|\mathbf{w}_j\|}$, i.e., the angle between $\mathbf{w}_i$ and $\mathbf{w}_j$. The models in the top row were trained with a larger learning rate (1e-3), and the bottom row a smaller learning rate (1e-4).

We also observed an interesting phenomenon: the orthogonality of the weights vanished during training. We tried to explicitly decorrelate the directions at initialization by introducing an orthogonal initializer (Saxe et al., 2014), but the orthogonality was still broken after training. It is interesting to see if the properties of initialization can be preserved in training, which is one of our future research directions.

## 3.4 DECISION BOUNDARIES IN A MANIFOLD REGION

Adversarial examples are benign inputs with imperceptible perturbations, which can dramatically degrade the performance of a machine learning model (Biggio et al., 2013; Szegedy et al., 2014). Adversarial attacks aim to push a benign input across the nearest decision boundary (Szegedy et al., 2014; Goodfellow et al., 2015; Moosavi-Dezfooli et al., 2016; Carlini & Wagner, 2017), hence it is important to study the decision boundaries in a manifold region.

The most important question we may ask is: *Do decision boundaries exist in a manifold region?* To address this question, we seek the point $\mathbf{x} \in S_L$ which has the maximum probability to be classified as Class $t \in \{1, 2, ..., M\}$, by solving the following optimization problem:

$$\max_{\mathbf{x}} \quad \mathbf{z}_t(\mathbf{x}) - \log\left(\sum_{j=1}^{M} \exp(\mathbf{z}_j(\mathbf{x}))\right) \tag{6}$$

$$\text{s. t.} \quad \mathbf{w}_i^T \mathbf{x} + b_i \geq 0, \quad \forall(\mathbf{w}_i, b_i) \in \mathcal{C}^*,$$
$$\text{MIN}_\mathbf{x} \leq \mathbf{x} \leq \text{MAX}_\mathbf{x},$$

where $M$ is the number of classes, and $\mathbf{z}_j(\mathbf{x})$ the $j$-th entry of the logits $\mathbf{z}(\mathbf{x})$. When $\mathbf{x}$ satisfies the constraints, i.e., $\mathbf{x} \in S_L$, $\mathbf{z} : S_L \to \mathbb{R}^M$ is an affine function of $\mathbf{x}$, hence $\mathbf{z}_j(\mathbf{x})$ can be rewritten as:

$$\mathbf{z}_j(\mathbf{x}) = (\nabla_\mathbf{x} \mathbf{z}_j(\mathbf{x}^*))^T (\mathbf{x} - \mathbf{x}^*) + \mathbf{z}_j(\mathbf{x}^*), \quad \forall \mathbf{x} \in S_L \tag{7}$$

which implies that the optimization objective in (6) is concave. The classification region of Class $t$ exists in $S_L$ if and only if $\mathbf{x}^t$ is classified into Class $t$.

$\mathbf{x}^t$ lies on the faces of the polytope in most cases, as shown in Appendix D. Since computing the volume of a polytope given by only the V-/H- representation is known as a #P-hard problem (Dyer & Frieze, 1988), the distance between $\mathbf{x}^t$ and the original test point $\mathbf{x}^*$, called *distortion*

in this paper, is used to informally measure the size of a linear region:

$$distortion = \max_{t \in \{1,2,...,M\}} \|\mathbf{x}^t - \mathbf{x}^*\|. \tag{8}$$

We randomly selected 1000 test points to search for the classification regions in their manifold regions. The average number of classification regions and the average distortions are shown in Table 2. The results show that a manifold region may contain several classification regions, which contradicts the conjecture in Hanin & Rolnick (2019) that two points falling into the same linear region are unlikely to be classified differently. We can also find that training with a large learning rate, using dropout or BN can make the manifold regions less likely to contain decision boundaries. However, this is not the case for BN in CNNs (see Appendix E for more information). In addition, BN can significantly decrease the distortion of $\mathbf{x}^t$, which implies a smaller size of the linear regions. Figure 5 shows a specific example of $\mathbf{x}^t$ generated from models trained with different optimization techniques, which also demonstrates our observation.

Table 2: Average number of classification regions in a manifold region, and the average distortions.

| Measure | Learning Rate | Vanilla | BN | Dropout |
|---|---|---|---|---|
| Number | 1e-3 | 1.116 | 1.083 | 1.012 |
| | 1e-4 | 8.766 | 2.622 | 1.048 |
| Distortion | 1e-3 | 26.35 | 14.90 | 25.28 |
| | 1e-4 | 23.81 | 07.61 | 25.41 |

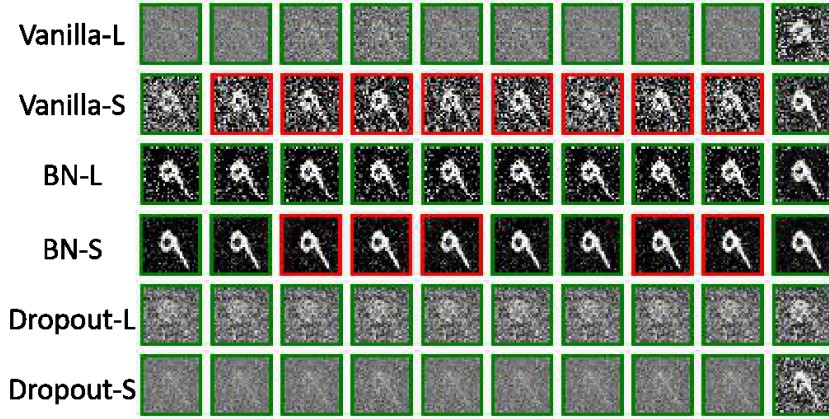

Figure 5: Different $\mathbf{x}^t$ of the point $\mathbf{x}^*$, which is classified into Class 10 (number $9$). From left to right, $t$ varies from 1 to 10 (number 0 to number 9). Images with green borders are correctly classified into Class 10, and red borders misclassified into Class $t$. Different rows represent the models trained with different optimization techniques or learning rates. The models whose names end with "-L" were trained with a larger learning rate (1e-3), and "-S" a smaller learning rate (1e-4).

Probing decision boundaries from the linear region perspective can bring many potential advantages for analysis. Due to the linearity of the manifold regions, high-order adversaries are not guaranteed to be better than first-order ones. Besides, adding random noise may help generate adversarial examples since more linear regions can be included during searching, which may explain why PGD with random restarts is a powerful adversarial attack approach (Madry et al., 2018).

## 3.5 RELEVANCE OF THE SURROUNDING REGIONS

The expressivity of a DNN depends on the number of the linear regions, but it makes optimization challenging when the gradient information of the surrounding regions has little relevance.

Given an example $\mathbf{x}^*$, we try to find an $\mathbf{x}$ from a different region, with $\mathbf{x} - \mathbf{x}^*$ parallel to all the decision boundaries of the region in which $\mathbf{x}^*$ lies, i.e., orthogonal to the directions of the decision

boundaries. Then we compare the gradient information between $\mathbf{x}$ and $\mathbf{x}^*$. As the classification result of $\mathbf{x}$ is equal to $\arg\max_{j=\{1,2,...,M\}} \mathbf{z}_j(\mathbf{x})$, the directions of the decision boundaries can be written in the following form:

$$\nabla_{\mathbf{x}}\left(\mathbf{z}_j(\mathbf{x}) - \mathbf{z}_k(\mathbf{x})\right), \quad j \neq k. \tag{9}$$

Therefore, to satisfy the orthogonality, we can sample an $L_2$-normalized direction $\mathbf{e}$ from the nullspace of the matrix $A = [\nabla_{\mathbf{x}}\mathbf{z}_1(\mathbf{x}^*), \nabla_{\mathbf{x}}\mathbf{z}_2(\mathbf{x}^*), ..., \nabla_{\mathbf{x}}\mathbf{z}_M(\mathbf{x}^*)]$:

$$N(A) = \{\mathbf{e} \in \mathbb{R}^d \mid A^T\mathbf{e} = 0, \|\mathbf{e}\| = 1\}, \tag{10}$$

and then $\mathbf{x}$ can be written as:

$$\mathbf{x} = \mathbf{x}^* + \beta\mathbf{e}, \quad \forall \beta \in [0, \epsilon], \mathbf{e} \in N(A), \tag{11}$$

so that

$$\left(\nabla_{\mathbf{x}}(\mathbf{z}_j(\mathbf{x}^*) - \mathbf{z}_k(\mathbf{x}^*))\right)^T (\mathbf{x} - \mathbf{x}^*) = 0. \tag{12}$$

Our experiments were performed on 1000 randomly selected test points. For each test point $\mathbf{x}^*$, we randomly sampled 100 $L_2$-normalized direction $\mathbf{e} \in N(A)$. As $\beta$ varied from 0 to $\epsilon$, $\mathbf{x}$ may belong to different regions. We searched for the linear regions along these directions ($\epsilon = 0.2$) and counted the number of unique linear regions. As shown in Figure 6(a), models using BN or trained with a smaller learning rate can usually partition the input space into more linear regions.

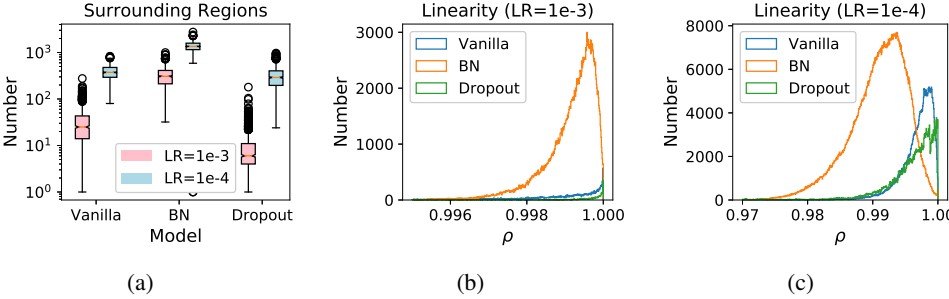

Figure 6: (a): boxplot of the number of unique surrounding regions for 1000 test points. Different colors represent different learning rates (LRs). (b) and (c): distributions of the cosine similarity $\rho$ of all unique surrounding regions. Models in (b) were trained with a larger learning rate (1e-3), and (c) a smaller learning rate (1e-4).

Next, we checked the relevance of these unique surrounding regions. First, we chose an $\mathbf{x} = \mathbf{x}^* + \beta\mathbf{e}$ from each unique linear region and calculated $\mathbf{g}(\mathbf{x})$ as its decision direction:

$$\mathbf{g}(\mathbf{x}) = \nabla_{\mathbf{x}}\left(\mathbf{z}_{k_1}(\mathbf{x}) - \mathbf{z}_{k_2}(\mathbf{x})\right), \tag{13}$$

where $k_1$ and $k_2$ are the indices of the two largest elements of logits. We calculated the cosine similarity $\rho = \frac{\mathbf{g}(\mathbf{x})^T\mathbf{g}(\mathbf{x}^*)}{\|\mathbf{g}(\mathbf{x})\|\|\mathbf{g}(\mathbf{x}^*)\|}$ to measure the relevance of different linear regions. As illustrated in Figures 6(b) and 6(c), a linear region has high relevance to its surrounding regions, which demonstrates the linearity of DNNs. It also shows that BN slightly reduces the relevance.

However, it was found that this relevance decreases rapidly with the depth of the DNN, resulting in gradients that resemble the white noise, which is known as the shattered gradients problem (Balduzzi et al., 2017). The low relevance also makes it hard to perform adversarial attacks because of the limited information of the local gradients (Athalye et al., 2018).

## 4  DISCUSSIONS AND CONCLUSIONS

This paper proposes novel tools to analyze the linear regions of DNNs. We explored the linear regions of models trained with different optimization settings, and observed significant differences of

their inspheres, directions of the hyperplanes, decision boundaries, and relevance of the surrounding regions. Our empirical observations illustrated that different optimization techniques may introduce different properties of the linear regions, even though they have similar classification accuracies. This phenomenon indicates that a more meticulous perspective is needed to study the behaviors of DNNs.

For DNNs with smooth nonlinear activations, it is still an open problem to define their linear regions with soft transition boundaries. One way to address the problem is to approximate the activation function by a piecewise linear function, in which case our approaches can directly be utilized to analyze the linear regions. By choosing a local linearity measure, like the one proposed by Qin et al. (2019), a soft linear region may also be precisely described by a set of inequalities, which require that the local linearity of each neuron is larger than a certain threshold. However, our approaches cannot be applied to these linear regions since their convexity is not guaranteed.

Our future research will reduce the high computational cost of our approach when applied to deeper and larger DNNs, and formulate a theoretical framework to explain why different optimization techniques can result in different properties of the linear regions. Eventually, we will try to answer the following questions:

- *What is the relationship between the properties of linear regions and the behaviors of a DNN?*
- *What kinds of linear regions can best approximate the data manifold?*
- *How to design novel optimization techniques to achieve such linear regions?*

## 5 ACKNOWLEDGMENTS

This research was supported by the National Natural Science Foundation of China under Grant 61873321 and Hubei Technology Innovation Platform under Grant 2019AEA171. The authors would like to thank Guang-He Lee from MIT for pointing out the connection of our paper to theirs.

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

## A  ILLUSTRATION ON A TOY DATASET

Figure 1 shows a two-dimensional slice of a high-dimensional input space, which may not be convincing enough to describe the properties of linear regions through the entire input space. To address the problem, we trained three models (vanilla, BN, dropout) on a two-dimensional toy dataset to see if the properties still remain.

Figure 7 visualizes the toy dataset we constructed, which consists of two separate spirals, representing two different classes. The vanilla model used here was a fully-connected DNN which consisted of three hidden layers with 10 rectified linear nodes in each. The BN model was the vanilla model with BN added after the pre-activations of each hidden layer, and the dropout model added dropout after the activations (the drop rate was 0.2). The models were trained with Adam optimizer (LR=1e-3, $\beta_1$=0.9, $\beta_2$=0.999).

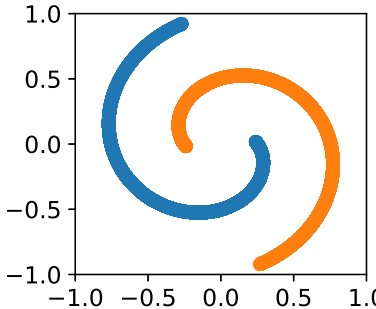

Figure 7: Visualization of the toy dataset. Different colors represent different classes.

The test accuracies for the three models were all 100%. Figure 8 plots linear regions and classification regions in the input space, illustrating the same properties as Figure 1 shows.

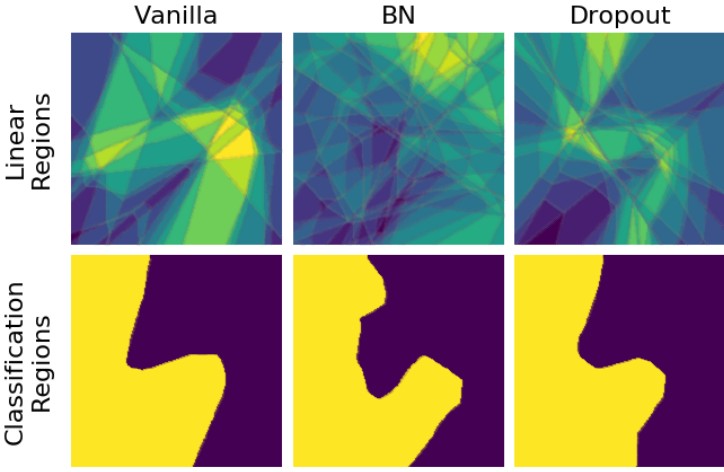

Figure 8: Linear regions and classification regions of models trained with different optimization techniques. The gray curves in the top row are transition boundaries separating different linear regions, and the color represents the activation rate of the corresponding linear region. In the bottom row, different colors represent different classification regions, separated by the decision boundary.

## B  CALCULATE PARAMETERS OF THE INEQUALITIES

As mentioned in Section 2.1, the first $l - 1$ hidden layers serve as an affine transformation of $\mathbf{x} \in S_{l-1}$. Besides, the pre-activation outputs of the $l$-th hidden layer $\mathbf{h}^l(\mathbf{x})$ are also an affine transformation of the activation outputs of the $(l - 1)$-th layer, hence $\mathbf{h}^l(\mathbf{x})$ is a linear function of $\mathbf{x} \in S_{l-1}$, which means $\mathbf{h}^l_n(\mathbf{x}) = \mathbf{w}^T_n \mathbf{x} + b_n$, where $n$ denotes a node of the $l$-th layer. So, we have:

$$\mathbf{w}_n = \nabla_{\mathbf{x}} \mathbf{h}^l_n(\mathbf{x}), \tag{14}$$

$$b_n = \mathbf{h}^l_n(\mathbf{x}) - \mathbf{w}^T_n \mathbf{x} = \mathbf{h}^l_n(\mathbf{x}) - (\nabla_{\mathbf{x}} \mathbf{h}^l_n(\mathbf{x}))^T \mathbf{x}. \tag{15}$$

Recall that according to the definition of $S_{l-1}$, $\mathbf{x}^*$ shares the same linear function as other $\mathbf{x} \in S_{l-1}$, therefore we have:

$$\mathbf{w}_n = \nabla_{\mathbf{x}} \mathbf{h}^l_n(\mathbf{x}^*), \tag{16}$$

$$b_n = \mathbf{h}^l_n(\mathbf{x}^*) - (\nabla_{\mathbf{x}} \mathbf{h}^l_n(\mathbf{x}^*))^T \mathbf{x}^*. \tag{17}$$

Then we should select $\mathbf{x}$ from $S_{l-1}$ to construct $S_l$, which means $\mathbf{x}$ satisfies $\mathrm{sgn}(\mathbf{h}^l_n(\mathbf{x}^*))\mathbf{h}^l_n(\mathbf{x}) \geq 0$, i.e., $\mathbf{x}$ has the same activation state of Node $n$ as $\mathbf{x}^*$. For convenience, $\mathbf{w}_n$ and $\mathbf{b}_n$ are directly multiplied by $\mathrm{sgn}(\mathbf{h}^l_n(\mathbf{x}^*))$ to obtain the form in (4).

## C  REMOVE REDUNDANT INEQUALITIES

Every constraint in $S_L$ represents a halfspace, but some of them are redundant, i.e., they are completely overrode by others. To eliminate the redundant constraints, we need to successively solve the following linear programming problem for each untested constraint and sequentially remove the redundant constraint from $\mathcal{C}^*$:

$$y = \min_{\mathbf{x}} \quad \mathbf{w}^T_k \mathbf{x} + b_k \tag{18}$$

$$\text{s. t.} \quad \mathbf{w}^T_i \mathbf{x} + b_i \geq 0, \quad \forall_{i \neq k} (\mathbf{w}_i, b_i) \in \mathcal{C}^*,$$

$$\mathbf{w}^T_k \mathbf{x} + b_k + 1 \geq 0,$$

$$\mathrm{MIN}_{\mathbf{x}} \leq \mathbf{x} \leq \mathrm{MAX}_{\mathbf{x}},$$

The constraint $\mathbf{w}^T_k \mathbf{x} + b_k \geq 0$ is redundant if and only if $y \geq 0$.

Note that each and every constraint needs to be tested sequentially, resulting in very heavy computational cost when dealing with thousands of constraints.

## D  $\mathbf{x}^t$ LIES ON THE FACES OF THE LINEAR REGION IN MOST CASES

**Proposition 1** $\mathbf{x}^t$ defined in Section 3.4 lies on the faces of its linear region in most cases.

**Proof** Let $M$ denote the number of classes, $\mathbf{x}^* \in \mathbb{R}^d$ a given point lying in the bounded linear region $S$, $\partial S$ the faces of $S$. Recall that the function fitted in a linear region is completely linear, hence the $j$-th logit $\mathbf{z}_j(x)$ can be rewritten as:

$$\mathbf{z}_j(\mathbf{x}) = \mathbf{w}^T_j \mathbf{x} + b_j, \quad \forall j \in \{1, 2, ..., M\}. \tag{19}$$

As defined in (6), let $p_t(\mathbf{x}) = \frac{\exp(\mathbf{w}^T_t \mathbf{x} + b_t)}{\sum_j \exp(\mathbf{w}^T_j \mathbf{x} + b_j)}$, then $\mathbf{x}^t = \arg\max_{\mathbf{x} \in S} \log p_t(\mathbf{x})$. If $\mathbf{x}^t \in S \setminus \partial S$, $\mathbf{x}^t$ must be a maximum, which means:

$$\nabla_{\mathbf{x}} \log p_t(\mathbf{x}^t) = \sum_{i=1}^M p_i(\mathbf{x}^t)(\mathbf{w}_t - \mathbf{w}_i) = 0. \tag{20}$$

Therefore,

$$\mathbf{w}_t = \sum_{i=1}^M p_i(\mathbf{x}^t)\mathbf{w}_i. \tag{21}$$

Recall that $\mathbf{w}_i \in \mathbb{R}^d$ and $d \gg M$, hence in most cases $\{\mathbf{w}_i\}_{i=1}^M$ are linear independent, which means $\mathbf{w}_t$ cannot be represented as the linear combination of $\{\mathbf{w}_{i \neq t}\}_{i=1}^M$. Since $0 < p_t(\mathbf{x}^t) < 1$, there is no $\{p_i(\mathbf{x}^t)\}_{i=1}^M$ satisfying (21). Therefore, $\mathbf{x}^t \in \partial S$.

# E   EXPERIMENTS ON THE CIFAR-10 DATASET

We also analyzed simple CNN models trained on the CIFAR-10 dataset. The results are presented in this Appendix.

## E.1   MODELS

The base architecture of the CNN models is shown in Table 3. To simplify the analysis, the convolutional layers were not padded, and large strides were used to replace the max-pooling layers. For the BN model, BN was added before ReLU activations. For the dropout model, dropout layers were added after ReLU activations.

Table 3: Architecture of the vanilla CNN model.

| Layers | Parameters | Activation |
|--------|------------|------------|
| Input | input size=(32, 32)$\times$3 | - |
| Conv | filters=(3, 3)$\times$32; strides=(2, 2) | ReLU |
| Conv | filters=(3, 3)$\times$64; strides=(2, 2) | ReLU |
| Conv | filters=(3, 3)$\times$128; strides=(2, 2) | ReLU |
| Flatten | - | - |
| Dense | nodes=1024 | ReLU |
| Dense | nodes=10 | Softmax |

The models were trained with Adam optimizer (learning rate 1e-3, $\beta_1$=0.9, $\beta_2$=0.999) on the CIFAR-10 dataset without data augumentation. The pixel values were rescaled to [-1, 1]. The batch size was 256, and early stopping was used to reduce overfitting. After training, the accuracies on the test set of the vanilla, BN and dropout models were 68.2%, 70.1% and 69.2%, respectively. Since the goal of this paper is to show that different optimization techniques can result in models with similar classification accuracies but significantly different linear regions, we only trained our models with one learning rate (1e-3), because other learning rates (1e-2, 1e-4) led to models with significantly different classification accuracies.

## E.2   PROPERTIES

As we have claimed in Section 2, a CNN model can be regarded as a highly sparse fully-connected DNN, which means we can analyze the properties of its linear regions in the same way. However, a convolutional layer may introduce some other properties into the linear regions. For example, considering the sparsity of the weights, a node of a convolutional layer can only partition a subspace into linear regions (instead of the whole input space, like a node in fully-connected DNNs does). Therefore, the partition of the linear regions are relatively independent for the nodes in a CNN, as most subspaces do not intersect. A direct consequence of their independency is that the properties of the linear regions in a CNN are more stable than those in a fully-connected DNN, which will be shown in the following experiments.

We randomly selected 1000 test points to perform the analysis. The inspheres of the linear regions in the CNNs are shown in Figure 9. Compared with the MNIST dataset, the CIFAR-10 dataset is more complicated and the mean point of a class is no longer classified into the same class, hence we randomly sampled a point from another class to replace the mean point described in Section 3 when constructing the decision regions. The experimental results show that both BN and dropout can narrow the linear regions. For dropout, the effect is more obvious on the decision boundaries than on the manifold regions.

Figure 10 shows the angles between different hyperplanes of a linear region. According to the architecture of the models, the first $15 \times 15 \times 32 + 7 \times 7 \times 64 + 3 \times 3 \times 128 = 11,488$ hyperplanes were provided by the convolutional layers, and the last $1,024$ by the fully-connected hidden layer. Since a node in a convolutional layer only partitions a certain subspace into regions, most of the pixel values are 0 in Figure 10 because there is no intersection between two subspaces. However, we can still observe that BN can orthogonalize the directions of the hyperplanes, whereas dropout the opposite.

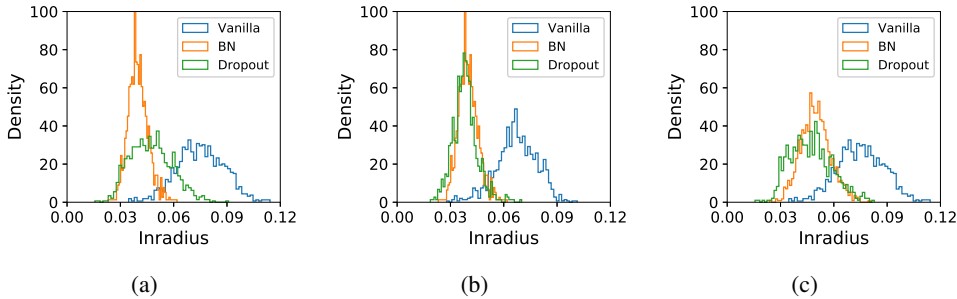

Figure 9: Inradius distributions of different optimization techniques for CNN models.(a) Manifold region; (b) Decision region; (c) Adversarial region.

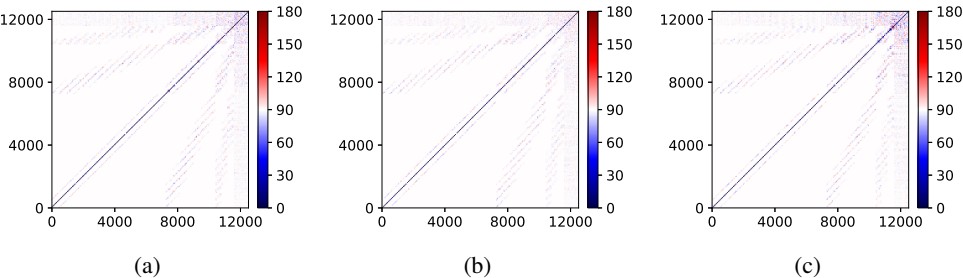

Figure 10: Angles between directions of different hyperplanes of a single manifold region. The indices on both axes indicate different hyperplanes: 0-7,199 represent the hyperplanes of the first convolutional layer, 7,200-10,335 the second convolutional layer, 10,336-11,487 the third convolutional layer, and 11,488-12,511 the fully-connected hidden layer. The value of a pixel is $\arccos \frac{\mathbf{w}_i^T \mathbf{w}_j}{\|\mathbf{w}_i\|\|\mathbf{w}_j\|}$, i.e., the angle between $\mathbf{w}_i$ and $\mathbf{w}_j$. (a) Vanilla; (b) BN; (c) Dropout.

Decision boundaries in the manifold regions of the CNNs seem less likely to be eliminated, compared with the fully-connected DNNs. As shown in Table 4, dropout can eliminate the decision boundaries in the manifold regions of CNNs, but BN did not work anymore. However, according to the distortions of the three models, BN still showed its ability to reduce the size of the manifold regions. Figure 11 shows a specific example of $\mathbf{x}^t$, whose original point was classified as *automobile*. $\mathbf{x}^t$ of the BN model had smaller distortion, which was also observed in fully-connected DNNs.

Table 4: Average numbers of the classification regions in a manifold region, and the average distortions.

| Measure | Vanilla | BN | Dropout |
|---|---|---|---|
| Number | 7.811 | 9.984 | 2.901 |
| Distortion | 27.08 | 20.85 | 25.25 |

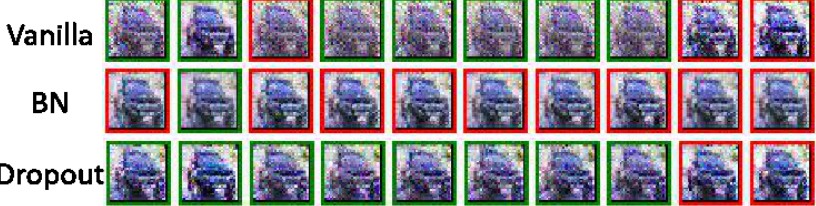

Figure 11: Different $\mathbf{x}^t$ of a point which is classified into Class 2 (automobile). From left to right, $t$ varies from 1 to 10 (airplane, automobile, bird, cat, deer, dog, frog, horse, ship, truck). Images with green borders are correctly classified into Class 2, and red borders misclassified into Class $t$. Different rows represent models trained with different optimization techniques.

The numbers of unique surrounding regions of a manifold region are shown in Figure 12(a). Though BN showed its strength in increasing the number of the linear regions, the effect was less obvious compared with those in Figure 6(a) for fully-connected DNNs. As we have pointed out before, the sparsity of the weights in the convolutional layers contributes to the stability of the properties of the linear regions. The relevance of the surrounding regions shown in Figure 12(b) was similar to fully-connected DNNs. BN still shattered the decision directions, as well as dropout, which was slightly different from the results for fully-connected DNNs.

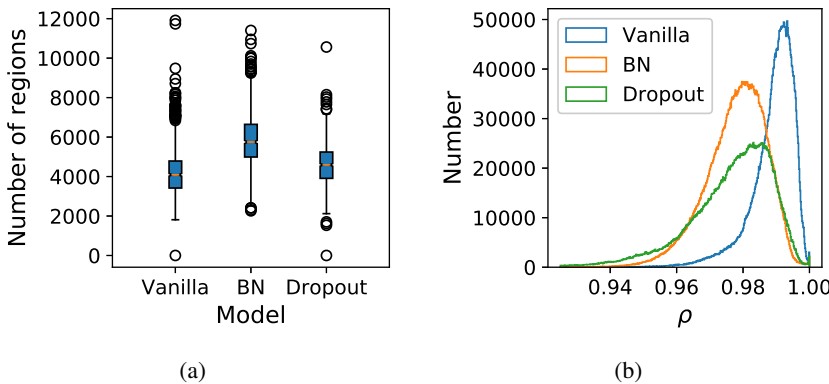

(a)  (b)

Figure 12: (a) Boxplot of the number of unique surrounding regions for 1000 test points; (b) Distributions of the cosine similarity $\rho$ of all unique surrounding regions for the 1000 test points.

