# OpenReview forum: "Empirical Studies on the Properties of Linear Regions in Deep Neural Networks"
_ICLR.cc/2020/Conference — Accept (Poster)_

### Official Review · AnonReviewer2 · 2019-10-21
**Official Blind Review #2**

**Rating:** 3

**Review:**

This work presents an array of analytical tools to characterize linear regions of deep neural networks (DNNs). Using the tools the work analyzes the effect of dropout and batch normalization (BN) on the linear regions of trained DNNs; namely, by assessing the properties such as inspheres, orientation of hyperplanes, decision boundaries and relevance of surrounding regions, the authors highlight the differences and similarities of linear regions induced by vanilla SGD as compared to SGD with dropout or BN.

The paper is clearly written and is easy to follow for the most part. The paper indeed presents a number properties for analyzing the nature of linear regions in DNNs; however it falls short of connecting them with an improvement in the optimization or interpretability of DNNs.

Even with respect to providing support for general applicability, the work does not go very far: without enough variation in data (not just image benchmarks), tasks and architecture, it is hard to determine if the analysis tools presented in the paper generalize beyond the chosen setup. For instance just the optimization techniques compared in the paper have their own hyperparameters and it is not clear how the results might vary with them.

I am not sure what to take away from figure 1 since it's only a two-dimensional slice of very high-dimensional input space. Maybe the authors could instead choose an example with a low-dimensional input space for illustration purposes.

Also, how much can be perceived from distributions shown in figure 2, since inradius (Eq. 5) may turnout to be a very coarse representation of linear regions, especially for deeper networks. Can the authors clarify this? Moreover, how would the figures look if we were using a different objective, dataset or architecture?

In figure 3, is it not possible to show the average results instead of just one example?

I would further like to know how the authors would deal with scalability issues if their analysis were to applied to more realistic (i.e. large) network architectures.

**Experience Assessment:**

I do not know much about this area.

**Review Assessment: Checking Correctness Of Derivations And Theory:**

I assessed the sensibility of the derivations and theory.

**Review Assessment: Checking Correctness Of Experiments:**

I assessed the sensibility of the experiments.

**Review Assessment: Thoroughness In Paper Reading:**

I read the paper at least twice and used my best judgement in assessing the paper.

---

> ### Author Response · Authors · 2019-11-14
> **Response to Reviewer #2 - part1**
>
> Thank you for the valuable feedback! We respond to the weakness in the following and hope we have addressed all the concerns.
>
> - The paper is clearly written and is easy to follow for the most part. The paper indeed presents a number properties for analyzing the nature of linear regions in DNNs; however it falls short of connecting them with an improvement in the optimization or interpretability of DNNs. Even with respect to providing support for general applicability, the work does not go very far: without enough variation in data (not just image benchmarks), tasks and architecture, it is hard to determine if the analysis tools presented in the paper generalize beyond the chosen setup. For instance just the optimization techniques compared in the paper have their own hyperparameters and it is not clear how the results might vary with them.
>
> The main purpose of our paper is to provide a new geometric perspective to study the linear region instead of just counting them. The number of the linear regions represents the expressivity of a DNN, but fails to indicate the influence of region’s geometric properties on some local behaviors of DNN, such as robustness. We think our research may give some new inspirations for studying linear regions. There are a number of studies on linear regions, and our experimental setting mostly followed a previous paper [1]. In addition to the fully-connected DNN, we also presented the results of a simple CNN on the CIFAR-10 dataset, which showed similar patterns. We also performed the experiment on a toy dataset as you suggested, whose results demonstrated similar properties of linear regions. As there are so many choices for training a DNN, we cannot fit all of them in a single paper; so, we put the emphasis on BN and dropout while keeping other hyper-parameters by default. However, some of our findings were also observed in other studies [2][4], which shows the generalization of our results beyond the chosen setup.
>
> - I am not sure what to take away from figure 1 since it's only a two-dimensional slice of very high-dimensional input space. Maybe the authors could instead choose an example with a low-dimensional input space for illustration purposes.
>
> The figure showing a two-dimensional slice of the input space was widely used in other papers to show some intuitive properties of the linear regions [1][2][3], and we thought it is suitable for illustration purposes. However, we do believe it is important to precisely illustrate the properties; so, according to your suggestion, we added another experiment on a toy 2D dataset in the revision. Please see Appendix A for more details.

---

> ### Author Response · Authors · 2019-11-14
> **Response to Reviewer #2 - part2**
>
> - Also, how much can be perceived from distributions shown in figure 2, since inradius (Eq. 5) may turnout to be a very coarse representation of linear regions, especially for deeper networks. Can the authors clarify this? Moreover, how would the figures look if we were using a different objective, dataset or architecture?
>
> As mentioned in Section 2.2, a polytope can be represented in V-representation or H-representation, but it is challenging to convert one representation into the other. If we want to explore the size of a linear region, V-representation would be a better choice. However, only H-representation can be obtained from the activation states of the DNN, resulting in difficulties in calculating the size. As a result, we used inradius to measure the narrowness of a linear region, which is related to its size and can be easily calculated from the H-representation. In our experiments, different optimization techniques did lead to different narrowness of the linear regions. In addition, the results of a simple CNN trained on the CIFAR-10 dataset, which was presented in Appendix E, showed similar patterns.
>
> - In figure 3, is it not possible to show the average results instead of just one example?
>
> The hyperplanes of different points are not in one-to-one correspondence, which means a pixel of one example has no relationship with the same pixel of another one, hence we think using average results may not be reasonable here.
>
> - I would further like to know how the authors would deal with scalability issues if their analysis were to applied to more realistic (i.e. large) network architectures.
>
> It is indeed a limitation of our approach when applied to large network architectures, as mentioned in Section 4. However, we have some preliminary thoughts on dealing with this problem. On the one hand, $\mathbf{x}^*$ is naturally an initial feasible solution of the convex optimization problem, which benefits the optimization process at the beginning. Moreover, large network architectures are usually CNNs, resulting in sparse weights of the inequalities, which may also help accelerate the optimization process. On the other hand, architectures used now are believed to be over parameterized, hence we may reduce the redundant parts before we analyze the architectures. However, these are just our preliminary thoughts, and a lot more studies are required.
>
> [1]Boris Hanin and David Rolnick. Complexity of linear regions in deep networks. ICML, 2019.
> [2]Novak et al. Sensitivity and generalization in neural networks: An empierical study. ICLR, 2018.
> [3]Boris Hanin and David Rolnick. Deep ReLU Networks Have Surprisingly Few Activation Patterns. NeurIPS, 2019
> [4]Balduzzi et al. The shattered gradients problem: If resnets are the answer, then what is the question? ICML, 2017.

---

### Official Review · AnonReviewer3 · 2019-10-21
**Official Blind Review #3**

**Rating:** 6

**Review:**

First, I believe that the acknowledgements in the manuscript give identifying information which could stand in conflict with a double blind review process. I’ll leave it to the area chairs/program chairs to make a decision on this. The following review will be contingent on the fact that the authors did not break the submission rules.

This paper aims to give new insights into deep neural networks by presenting a number of approaches to analyse the linear regions in such networks. The authors define a linear region around a point x* as the intersection between a number of half spaces that are defined through linear approximations of a DNN (tangents) around that point x*. The authors show that points within these regions can be found using convex optimization with a number of linear constraints that are equal or less than the number of nodes in a DNN. In experiments with a fully connected network the authors analyse different properties of these linear regions: (1) How big is the biggest sphere that we can fit in a linear region? (2) How much do the hyperplanes that define a region correlate with each other? (3) How reliably does a linear region represent a single class? And (4), How does a linear region interact with neighbouring regions? In their presentation, the authors focus on comparing these properties between models that were either trained without regularisation, with batch normalisation, or with dropout and with different learning rates. This allows them to draw insightful conclusions about the difference between linear regions in these models. The authors hope that their work will enable new ways of analysing DDNs that will inspire new architectures and optimization techniques.

I vote to accept this paper. The authors present a large array of methods to analyse the linear regions of DNNs. Their insights into the differences of BN and Dropout are useful (figure 1 & 2) and sensible (figure 3). The implications of linear regions on adversarial robustness can have an impact in the future. Because the paper relies on geometrical reasoning, I wished there would be more visualisations that guide the reader.

Here are a number of comments and questions that I have on the manuscript:
- Figure 1 Top: What do the different colour represent in the linear regions plot?
- Section 2.1: maybe add a toy graph that visualises the depth-wise ‘exclusion’ process of feasible “neighbours” of x*?
- Eq. (2) & (3): Explain where these equations come from.
- Sec 3.2, first sentence. The authors claim that inspheres of linear regions are highly relate to the expressivity of DNN. Can they elaborate on that claim? Is this claim a result of their experiments?
- What is the relationship between the number of constraints in eq. (5) and the radius of an insphere? Does the insphere size decrease with more constraints? What implications would that have on deeper networks than the one that was presented?
- Why should distortion be a good measure of the size of a linear region?
- The authors claim that it is expensive to run their approach, and that they will aim to improve speed in the future. Can the authors give a more concrete example of runtimes in their current approach?


**Experience Assessment:**

I do not know much about this area.

**Review Assessment: Checking Correctness Of Derivations And Theory:**

I assessed the sensibility of the derivations and theory.

**Review Assessment: Checking Correctness Of Experiments:**

I carefully checked the experiments.

**Review Assessment: Thoroughness In Paper Reading:**

I read the paper thoroughly.

---

> ### Author Response · Authors · 2019-11-14
> **Response to Reviewer #3 - part1**
>
> We thank the reviewer for the constructive comments, which helped improve the paper. We also apologize for our mistake for including the acknowledgement. It has been removed in our revision. Thank you for pointing this out!
> We address your detailed comments below.
>
> - Figure 1 Top: What do the different colour represent in the linear regions plot?
>
> The color represents the ratio of the activated nodes in a linear region. We added this in the caption of Fig. 1 in the revision. Albeit that different colors were used to separate linear regions in the previous paper, we believe our plot can provide more information since the gray lines have already illustrated in different regions.
>
> - Section 2.1: maybe add a toy graph that visualises the depth-wise ‘exclusion’ process of feasible “neighbours” of x*?
>
> Thanks for your suggestion! We added Fig. 2 to illustrate this more clearly. Please check Section 2.1 in our revision.
>
> - Eq. (2) & (3): Explain where these equations come from.
>
> As mentioned in Section 2.1, the first $l-1$ hidden layers serve as an affine transformation of $\mathbf{x}\in S_{l-1}$. Besides, the pre-activation outputs of the $l$-th hidden layer $\mathbf{h}^l(\mathbf{x})$ are also an affine transformation of the activation outputs of the $(l-1)$-th layer, hence $\mathbf{h}^l(\mathbf{x})$ is a linear function of $\mathbf{x}\in S_{l-1}$, which means $\mathbf{h}^l_n(\mathbf{x})=\mathbf{w}_n^T\mathbf{x}+b_n$, where $n$ denotes a node of the $l$-th layer. For a linear function $y=\mathbf{w}^T\mathbf{x}+b$, the $\mathbf{w}$ can be directly calculated by $\mathbf{w}=\nabla_{\mathbf{x}}y$, whereas $b=y-\mathbf{w}^T\mathbf{x}$. Here $\mathbf{x}$ and $y$ can be replaced by $\mathbf{x}^*$ and $\mathbf{h}_n^l(\mathbf{x}^*)$ because $\mathbf{x}^*$ shares the same linear function as other $\mathbf{x}\in S_{l-1}$. Last, the parameters are multiplied by $\mbox{sgn}(\mathbf{h}_n^l(\mathbf{x}^*))$ to make sure that the inequalities, which indicate the activation states of the $l$-th layer, are all in the $\geq$ form.
> A formal deduction was added in Appendix B.
>
> - Sec 3.2, first sentence. The authors claim that inspheres of linear regions are highly relate to the expressivity of DNN. Can they elaborate on that claim? Is this claim a result of their experiments?
>
> It is believed that a DNN with more linear regions has a larger potential to fit complex functions [1][2]. For example, a regular hexagon is a better approximation of a circle than a square. Small inspheres do demonstrate the narrowness of the linear regions, resulting in a large number of regions.
> [1] Poole et al. Exponential expressivity in deep neural networks through transient chaos. NIPS, 2016.
> [2] Pascanu et al. On the number of response regions of deep feed forward networks with piece-wise linear activations. https://arxiv.org/abs/1312.6098. 2014
>
> - What is the relationship between the number of constraints in eq. (5) and the radius of an insphere? Does the insphere size decrease with more constraints? What implications would that have on deeper networks than the one that was presented?
>
> Yes, the radius does decrease with more constraints, or more precisely, irredundant constraints (which means the constraints cannot be implied by others). A smaller inradius usually results in a larger number of linear regions, hence deeper networks usually have higher fitting ability. Regarding the number of linear regions, i.e. the complexity of DNNs, a well-known question is that why deeper networks have better generalization, instead of overfitting? We believe it comes from the relevance among the linear regions. A node has a set of fixed weights, creating different constraints for different activation states of the preceding layers, which can be regarded as that part of the weights are picked to construct a constraint. However, different parts of the weights are chosen from the same set, resulting in this relevance. Maybe we are a little off the topic here, but it is really an interesting research direction. Another interesting direction is to show that depth provides irredundant constraints more efficiently than width, which is still our work in progress.

---

> ### Author Response · Authors · 2019-11-14
> **Response to Reviewer #3 - part2**
>
> - Why should distortion be a good measure of the size of a linear region?
>
> The best measure here should be the volume, but as mentioned in Section 3.4, calculating the volume of a high-dimensional polytope is really challenging. Though the inradius can show the narrowness of the linear region, it cannot represent the size of a linear region completely (just imagine a long rectangle), hence the exradius is also needed to describe the size. Unfortunately, calculating the exradius is also a difficult task for H-representation (though easy for V-representation), so we have to use distortion as a rough measure. $\mathbf{x}^t$ lies on the surfaces of the linear region, and is usually far from the decision boundaries. Let’s imagine a simple linear model: $\mathbf{x}^t$, whcih is the point with the highest probability to be classified as $t$, must be the farthest point to the decision boundary. Therefore, here we used distortion to roughly represent the exradius of a linear region.
>
> - The authors claim that it is expensive to run their approach, and that they will aim to improve speed in the future. Can the authors give a more concrete example of runtimes in their current approach?
>
> The most time-consuming part of our experiments is finding $\mathbf{x}^t$ in Section 3.4, costing about 37 seconds per sample. The convex optimization can be solved in polynomial time, depending on the optimizer used (here we use MOSEK https://www.mosek.com/), but the computing time increases with the number of the constraints and the input dimensionality. The optimization for each sample can run in parallel, but our computing resources were really limited.

---

### Official Review · AnonReviewer1 · 2019-10-24
**Official Blind Review #1**

**Rating:** 8

**Review:**

This paper addresses the following: how do batch normalization and dropout affect the number of linear regions present in a deep network? It does so by devising a search procedure for enumerating a set of linear inequalities that define the linear region around a particular input. The linear region is defined as the region of input space that activates the same units/nodes in the network. The authors compute these linear regions for three different types of fully connected networks trained with: vanilla (nothing added), batch norm, and dropout. Given these linear regions, the authors studied a number of their properties, such as the radii of inscribed spheres, angles between hyperplanes, and number of unique surrounding regions.

Comments
- This paper enumerates a number of interesting findings, all of which seem to raise intriguing questions about the properties of trained networks. However, after reading the paper, I am left a little unsure of what to make of the results. However, I do not think this is a fault of the paper, instead I enjoyed that this paper raises so many interesting questions. Still, some more discussion and interpretation of the results is perhaps warranted.
- I especially enjoyed the writing, the problem statement and exposition were clear and easy to follow.
- Perhaps the authors could comment, in the discussion, if they think their methods could be extended to networks with smooth nonlinearities (such as tanh), or what aspects of their results are also apply to networks with different nonlinearities.
- I was also curious if the authors could comment on similarities and differences between their findings and this relevant paper (https://arxiv.org/abs/1802.08760) by Novak et al. that empirically computes linear regions for 2D slices through input space.

Minor edits
- After introducing the definition of the insphere (eq 5), it would be helpful to remind the reader that this is for a particular region defined by the set of inequalities C^\*.
- Typo in footnote 2 on page 5: partitioned

**Experience Assessment:**

I have read many papers in this area.

**Review Assessment: Checking Correctness Of Derivations And Theory:**

I carefully checked the derivations and theory.

**Review Assessment: Checking Correctness Of Experiments:**

I assessed the sensibility of the experiments.

**Review Assessment: Thoroughness In Paper Reading:**

I read the paper at least twice and used my best judgement in assessing the paper.

---

> ### Author Response · Authors · 2019-11-14
> **Response to Reviewer #1**
>
> Thank you for the positive comments! Our detailed replies are given below.
>
> - This paper enumerates a number of interesting findings, all of which seem to raise intriguing questions about the properties of trained networks. However, after reading the paper, I am left a little unsure of what to make of the results. However, I do not think this is a fault of the paper, instead I enjoyed that this paper raises so many interesting questions. Still, some more discussion and interpretation of the results is perhaps warranted. I especially enjoyed the writing, the problem statement and exposition were clear and easy to follow.
>
> Thanks for your interest! We expanded our discussion according to your suggestions. Please see our revision for more details.
>
> - Perhaps the authors could comment, in the discussion, if they think their methods could be extended to networks with smooth nonlinearities (such as tanh), or what aspects of their results are also apply to networks with different nonlinearities.
>
> ‘Hard’ linear regions can only be defined when the activation is piecewise linear. However, we believe our findings can be extended to networks with smooth nonlinear activation, because a smooth nonlinear activation, like tanh, can be approximated by piecewise linear functions. So far there is no precise definition of ‘soft’ linear regions for DNNs with smooth nonlinearities, but we may provide some preliminary ideas to find these ‘soft’ linear regions. First, we need a local linearity measure, such as Eq. (5) in https://arxiv.org/abs/1907.02610; and then set a threshold of nonlinearity of every neuron, resulting in a set of inequalities to describe a ‘soft’ linear region. Unfortunately, our methods cannot be directly applied to analyzing these ‘soft’ linear regions since their convexity is not guaranteed. It is still an open problem to precisely analyze these ‘soft’ linear regions.
> A similar discussion was added in Section 4. Please check our revision for more information.
>
> - I was also curious if the authors could comment on similarities and differences between their findings and this relevant paper (https://arxiv.org/abs/1802.08760) by Novak et al. that empirically computes linear regions for 2D slices through input space.
>
> This is a highly related work to ours. Thanks for pointing this out! Fig. 3 in Novak et al. illustrates that the on-manifold regions are usually larger than the off-manifold regions after training, which is also implied by our Fig. 3: the manifold regions are usually larger than the decision regions (see the blue lines of the first two columns). Besides, our paper also shows some other properties of the linear regions, and compares the influences introduced by different optimization techniques.
> We updated Section 3.2 in our paper to include this discussion.
>
> - Minor edits
>
> We revised our manuscript according to your suggestions. Thanks again!

---

### Public Comment · ~Haokun_Luo1 · 2019-10-11
**Interesting work!**

Interesting work really, and it seems to be the only work on linear regions this time (;D). It is inspiring to study linear regions from geometric perspectives, instead of just counting the number, because some very local behaviors of DNNs, such as adversarial examples, are highly related to the properties of the linear region. However, I have some little comments here.

1. I’d like to mention a closely related paper [1], which also discussed the connection between linear regions and adversarial examples.
[1] B. Hanin and D. Rolnick, “Complexity of Linear Regions in Deep Networks,” in Proc. 36th Int’l Conf. on Machine Learning, Long Beach, CA, 2019, pp. 2596–2604.

2. In Section 3.4, the authors claimed that the “high-order adversaries are not guaranteed to be better than first-order ones”. It would be nice if the authors could give an example to make it clear.

---

> ### Author Response · Authors · 2019-10-12
> **Thanks for your comments**
>
> Thanks for your comments! Our detailed responses are as follows:
>
> 1.The paper you mentioned is very valueable, but it seems that only a heuristic conjecture, which discussed linear regions and adversarial examples, is presented in Section 2.2 (their work):
>
> “Moreover, the distance from a typical point to the transition boundaries of linear regions gives a heuristic lower bound for the typical distance to an adversarial example: two inputs closer than the typical distance to a linear region boundary likely fall into the same linear region, and hence
> are unlikely to be classified differently.”
>
> However, it is not consistent with what we observed. As the results we presented in Table 5, a linear region can also contain many classification regions, which means that you could find points with different labels in a linear region. Therefore, according to our experiments, we think the conjecture in [1] is arguable.
>
> The work you mentioned is highly related to our topic, so we will add it into reference in the revision. Again, thanks for pointing this out.
>
> 2.For second comment, you can consider the C&W loss function. The high-order derivatives of the C&W loss function with respect to the input are all 0, since the DNN behaves completely linear in the linear region. We will add this simple discussion in the revision to make our point clear.

---

> > ### Public Comment · ~Haokun_Luo1 · 2019-10-13
> > **Thanks a lot!**
> >
> > Thanks for author's quick response!

---

### Public Comment · ~Thiago_Serra1 · 2019-10-11
**Regarding depth and the number of linear regions**

It is great to see linear regions analyzed through new angles. As someone who has worked on the topic, I would like to add something to your discussion on literature review.

Regarding the comment that "Studies have shown that the number of the linear regions increases more quickly with the depth of the DNN than the width", there is actually an analytical  trade-off between depth and width that depends on the number of neurons and the size of the input: https://arxiv.org/abs/1711.02114

In Figure 5 of the mentioned paper, you can see that the maximum number of linear regions attainable by neural networks with 60 units according to the size of the input. The bound is exact for shallow networks (the case of 1 layer with 60 neurons), which implies that shallow networks may define more linear regions than deep networks for the same number of units if the size of the input is sufficiently large.

---

> ### Author Response · Authors · 2019-10-16
> **RE: Regarding depth and the number of linear regions**
>
> Thank you for your interest and additional information! It’s a nice work which achieved tighter bounds on the maximal number of linear regions and presented more detailed influence of the depth and width of DNNs. We will add this part of discussion into our Introduction. By the way, we think it also interesting to analyze the properties of linear regions introduced by the depth and width. However, there are so many details and choices for training a DNN and we cannot analyze them all, so we put the emphasis on BN and dropout in our paper.

---

> > ### Public Comment · ~Thiago_Serra1 · 2019-10-16
> > **Follow-up**
> >
> > Indeed, there is a lot to be analyzed and for sure you cannot fit it all in a single paper!

---

### Public Comment · ~Runyao_Chen1 · 2019-10-21
**About BN and adversarial examples**

It’s a nice work which may inspire other researchers!

The result in Table 5 shows that BN introduces smaller size of linear regions, but dose not reduce the number of classification regions in a linear region. It reminds me of another paper which claims that BN may be one of the causes of adversarial examples.

Galloway, Angus, et al. "Batch Normalization is a Cause of Adversarial Vulnerability." arXiv preprint arXiv:1905.02161 (2019). https://arxiv.org/abs/1905.02161

P.S. I think there is no need to present Table 1 because the architecture has already been clarified in the context.

---

> ### Author Response · Authors · 2019-10-24
> **Indeed!**
>
> Our observations do imply the same result: BN may be one of the reasons which leads to the vulnerability of DNNs. However, there are still some differences.
>
> Our results empirically showed that BN usually introduces smaller size of linear regions, but the number of classification regions in a linear region doesn't decrease along with the size, resulting in less robustness.
>
> Their observations showed that BN can lead to the tilting angles of the decision boundary w.r.t. the nearest-centroid classifier, especially when the variances of some hidden outputs are very small. As a result, many points are very close to the decision boundaries, resulting the vulnerability. However, their results cannot imply that BN leads to smaller size of linear regions whereas keeping the number of classification regions in a linear region nearly the same.
>
> By the way, we will delete Table 1 for brevity. Thanks for pointing this out!

---

### Decision · Program_Chairs · 2019-12-19

**Decision:**

Accept (Poster)

**Comment:**

This paper studies the properties of regions where a DNN with piecewise linear activations behaves linearly. They develop a variety of techniques to chracterize properties and show how these properties correlate with various parameters of the network architecture and training method.

The reviewers were in consensus on the quality of the paper: The paper is well written and contains a number of insights that would be of broad interest to the deep learning community.

I therefore recommend acceptance.